# A Load-Carrier Perspective Method for Evaluating Land Resources Carrying Capacity

**DOI:** 10.3390/ijerph19095503

**Published:** 2022-05-01

**Authors:** Wenzhu Luo, Liyin Shen, Lingyu Zhang, Xia Liao, Conghui Meng, Chi Jin

**Affiliations:** 1School of Economics and Management, Chongqing University of Posts and Telecommunications, Chongqing 400065, China; luowz@cqupt.edu.cn; 2International Research Centre for Sustainable Built Environment, School of Management Science and Real Estate, Chongqing University, Chongqing 400044, China; zhanglyxx@163.com; 3School of Territorial Space Planning, Zhejiang University City College, Hangzhou 310015, China; 4Institute of Geographic Sciences and Natural Resources Research, Chinese Academy of Sciences, Beijing 100045, China; liaoxia8289@igsnrr.ac.cn; 5School of Public Affairs, Zhejiang University, Hangzhou 310027, China; mengconghui@zju.edu.cn; 6Faculty of Architecture and the Built Environment, Delft University of Technology, 2629 HS Delft, The Netherlands; c.jin-1@tudelft.nl

**Keywords:** evaluation, land resources carrying capacity (LRCC), carrier and load perspective, multifunctional land use

## Abstract

If land resources are forced to withstand greater populations than they are able to withstand, irreversible damage to the land resources system will happen in a specific region. This challenge highlights the urgency of appropriately evaluating the land resources carrying capacity (LRCC). A proper level of the capacity can ensure that land resources demands imposed by human activities are at a reasonable level. There is a need for a proper evaluation method for assessing LRCC. This study presents a new evaluation method from a load-carrier perspective for assessing LRCC by examining the relationships between the pressure caused by human activities and the supply capacity of land resources. In developing this method, a land resources system is determined by two primary components, namely carrier and load. The compositions of carrier and load are determined by applying the theory of multifunctional land use. A case demonstration is conducted to show the application of the method. The main findings can be drawn from this study as follows. Firstly, a “load-carrier” perspective method is requested for evaluating the regional LRCC, and it is effective in obtaining the value of LRCC in the demonstration case. Secondly, the composition of land resources carriers and loads embodied in the load-carrier perspective method is determined by using the theory of multifunctional land use. Thirdly, the case results suggest that seven regions are overloaded in LRCC and the other two regions are approaching the limitation of LRCC among nine county-level administration regions in Chongqing. This study contributes to the development of literature in the field of LRCC. The application of the “load-carrier” perspective method can help local governments in the case study regions make policies to ensure that land resources demands imposed by human activities are under control at a reasonable level.

## 1. Introduction

The demands of human socio-economic activities for land resources are exceeding the supply capacity of land resources. Previous studies [1,2] appreciate that land resources have already carried more population than they can do. Irreversible damage to the land resources system occurs, and ecological problems are becoming increasingly serious. Examples include the generation of CO_2_ caused by the consumption of fossil fuels, particularly in industrial activities; On the other hand, human demands for agricultural products have led to the expansion of cultivated land and grass land, resulting in the decrease of forests that can absorb CO_2_. The two reasons mentioned above lead to the increase of CO_2_, which contributes to global warming [3]. It is appreciated that overgrazing, deforestation, fuelwood consumption, the need to expand construction land, and industrial activities such as mining have led to the decrease in soil quality, namely soil degradation. In turn, soil degradation reduces land resources’ ability in supporting human living [4,5]. Therefore, it is imperative to evaluate land resources carrying capacity (LRCC).

The inherent perception of LRCC is that the larger scale or the better quality of land resources in a region, the larger the human socio-economic system that can be supported [6,7]. However, taking Hong Kong as an example, it has small scale of land resources and high per capita consumption of land resources. According to the inherent perception, the LRCC in Hong Kong is only comparable to a small or medium-sized city in the central or western regions of China. However, the land resources of Hong Kong support a population of 7.5 million. It can be seen that LRCC is also influenced by the human socio-economic system [8]. Therefore, it is necessary to evaluate LRCC by integrating the pressure imposed by human activities and the supply capacity of land resources. In this way, the evaluation results can help government manage land resources and regulate socio-economic activities, and keep the pressure caused by human activities within the supply capacity of land resources. Consequently, sustainable land resources use can be achieved. As shown by Tiffen and Mortimore [9], even in a densely populated region, if human activities are controlled at an appropriate level and effective land resources management measures are adopted, the carrying capacity of land resources can be adequately achieved, and thus soil degradation can be prevented.

According to previous studies [10,11,12], LRCC can be classified into three types: (i) the regional carrying threshold, (ii) the regional carrying status, and (iii) the global safe operating space. LRCC from the perspective of regional carrying threshold can be gained by using two methods, namely, based on limiting food or based on a reference region. LRCC based on limiting food is one of the earliest paradigms, which extends the theory introduced by Malthus [13] about the discussion of the human-food relationship, and focuses on the carrying threshold expressed in terms of population size [14]. For example, Hao et al. [15] investigated how large of a population could be supported by nine types of agricultural products (including grain, vegetable oil, sugar, fruits, meat, poultry eggs, milk, and fishery products) produced on cultivated land and grass land based on the daily calories, protein and fat of regular human needs. It can be seen that the limiting food based paradigm follows the research path of “land→population”, and a reference region based paradigm also follows this research path, as seen in the study by Sun and Liu [16] and Zhou et al. [17]. LRCC based on a reference region calculates the LRCC of a study area depending on per capita possession of land resources in a reference area and the stock of land resources in the study area.

The concept of LRCC from the perspective of regional carrying status is based on the ecological footprint and other factors. LRCC based on an ecological footprint focuses on the pressure of human activities on land resources, and compares the pressure with the supply capacity of land resources in order to obtain the result of surplus or deficit carrying status of land resources. This pressure-supply two-dimensional ecological footprint model was proposed by Wackernagel and Rees [18]. In this paradigm, land resources are taken as an important ecological dimension, and the research path is shifted from “land→population” to “population→land”. By employing the two-dimensional ecological footprint model, Peng et al. [19] conducted an empirical study in the context of Jiangsu Province during the period of 2012–2017. They argued that per capita ecological carrying capacity of Jiangsu Province increased during 2012–2017 but the pressure on cultivated land exceeded the supply capacity of cultivated land, showing a large deficit carrying status. Galli et al. [20] studied the LRCC in six metropolitan areas (including Almada, Bragança, Castelo Branco, Vila Nova de Gaia, Guimarães and Lagoa) in Portugal for the years of 2011–2016 based on the two-dimensional ecological footprint model. The results show that the per capita ecological footprint in these six metropolitan areas in 2016 was 4.01, 4.02, 3.66, 3.25 and 3.94 gha/person, respectively, which are all higher than the international average value on the per capita ecological footprint of 1.7 gha/person in 2016. On the other hand, LRCC from the perspective of regional carrying status based on multi-factors follows the research path of “land→population”, which integrates various factors affecting LRCC, as shown in studies [21,22,23]. However, the result from using a multi-factor research path is considered to not be helpful for a government’s policy planning [11].

Furthermore, LRCC from the perspective of the global safe operating space is based on a planetary boundaries framework. The concept of planetary boundaries originated from whether humans have affected the functioning of the Earth’s complex system at the global scale [24]. The Earth’s complex system has multiple stable status, when human disturbance breaks through the critical threshold, and the Earth’s complex system will undergo a critical transition, namely entering the next stable status [25,26]. As the Earth’s complex system enters a new stable status, humanity needs to adapt to it. The planetary boundaries are the maximum regenerative and absorptive capacity of the Earth’s complex system that can be safely occupied by humans, and are the critical threshold when the Earth’s complex system maintains the stability of its own structure and function [27]. Rockstrom et al. [28] defined nine key biophysical processes in the Earth’s complex system including climate change, ocean acidification, stratospheric ozone depletion, biogeochemical flow, freshwater use, land system change, rate of biodiversity loss, atmospheric aerosol loading, and chemical pollution. They set the safe operating space for the first seven processes mentioned above and measured the current status. The safe operating space mentioned above is taken at the initial end of the uncertain interval that is estimated from statistical analysis based on historical data [29], and thus it can be seen that the safe operating space is a relatively conservative estimate of the critical threshold. Rockstrom et al. [28] further used the proportion of cultivated land as an evaluation indicator to measure land system change, and found that the safe operating space is 15% and the current value is 11.7%. That is the LRCC based on planetary boundaries framework as understood in this study. Some studies [30,31,32] have extended the planetary boundaries framework at the global level to exploring LRCC at the regional level.

The above discussions show some research gaps left in the existing studies. Firstly, the carrying threshold for defining regional LRCC based on limiting food or a reference region is considered to not be correct. This is because that LRCC based on limiting food only considers the food supply function of land resources, but land resources have more functions than just food supply. At the same time, it is not reasonable to consider food as a limiting factor at small regional scales (e.g., municipality and county) with the help of food trade. On the other hand, LRCC based on a reference region considers the supply capacity of land resources in a study region but the demand for land resources in a reference region. However, it has not considered the differences between the reference region and the study region in terms of the basic condition and utilization efficiency of land resources. Therefore, the validity of using the demand for land resources in a reference area to measure LRCC in the study area is of concern. Secondly, the weakness of exploration of the regional carrying status is considered. It is appreciated that LRCC based on multi-factors integrates human activities and land resources, but it fails to analyze the relationship between human activities and land resources in the process of selecting indicators. This may obscure some information reflected in key indicators. And it cannot be applied at a specific time point for a specific region. It must refer to a period of time or a group of regions. On the other hand, LRCC based on ecological footprint integrates the pressure of human activities on land resources and the supply capacity of land resources, but the result cannot define the regional carrying threshold, and can only present the regional surplus or deficit carrying status. Thirdly, LRCC based on a planetary boundaries framework focuses on the safe operating space and current value of land system change at the global scale. However, this method fails to focus on the pressure and the supply capacity of land resources at small regional scales (e.g., municipality and county). In line with the above discussions on the weakness of existing in previous studies, this paper aims to propose a load-carrier perspective method for evaluating LRCC through examining the relationships between the pressure caused by human activities and the supply capacity of land resources. The method will be applicable to different regional scales, namely national, province, municipality, or county. The evaluation result by adopting this new method can answer the number of populations that can be carried by land resources at a specific time point for a specific region, and thus can help governments formulate measures to ensure that demands for land resources from human activities are under control at a reasonable level. Consequently, sustainable land resources use can be achieved.

The rest of the paper is structured as follows. Section 2 defines LRCC from a load-carrier perspective, and classifies land resources carriers and loads respectively. Subsequently, indicators for measuring land resources carriers and loads are selected and the measurement model for LRCC is established in Section 3, followed by a case demonstration for the application of the introduced method in Section 4. Section 5 presents discussions on the introduced method and demonstration results. And Section 6 summarizes the main findings and contributions of this study, and also appreciates the limitations as well as the future research recommendations.

## 2. Definitions of Carriers and Loads in Describing Land Resources Carrying Capacity

This paper refers to the definition of land resources shown in FAO [33], which means all natural environments that affect the potential land use such as climate, landforms, soils, vegetation, water, and the results of human activities. Within the land resources system, humans are able to continuously obtain goods (e.g., agricultural products) and enjoy the system’s services (e.g., climate regulation) if they do not destroy the system. On the other hand, resilience provides the land resources system with the ability to maintain its vitality in the face of human-induced disturbances, and the system maintains the existing structure and functions [34]. However, once the critical threshold of the land resources system is breached because of human-induced disturbances, the system will transform to others that appear with different structure and functions [35]. The critical threshold of the land resources system mentioned above can be expressed in terms of the physical carrying capacity, which means that the maximum load that can be carried by a physical material or member, namely carrier, without physical damage.

From the discussion above, LRCC must involve both carriers and loads, and this viewpoint is verified in the resources environment carrying capacity evaluation model introduced by Shen et al. [36], in the water resource carrying capacity assessment model presented by Liao et al. [37], and in the urban infrastructure carrying capacity presented in Wang et al. [38]. In this paper, LRCC is defined as the maximum number of populations that can be carried by land resources without the transformation of the land resources system at a specific time point for a specific region based on a load-carrier perspective. It can be seen that the land resources carriers are various functions that the land resources system can supply, and the land resources loads are various pressures caused by human activities. The relationships between LRCC, land resources carriers and loads are shown in Figure 1.

### 2.1. Land Resources Carriers

Li and Fang [39] introduced a theory of multifunctional land use and a comprehensive classification of land resources carriers as shown in Table 1, which is adapted from previous studies. Three typical classifications on land resources carriers are given in previous studies, namely ecosystem services [40,41,42,43,44], multi-functional agriculture [45,46], and multi-functional landscapes [47,48]. The classifications in Table 1 will be adopted in this study.

In referring to Table 1, certain land resources carriers will be either deleted or integrated in order to achieve the research objective. First, the five types of land resources carriers, including medicinal resources supplies, timber production, fiber production, the provision of energy resources, and minerals production are the foundation of industrial product supply; in other words, the five resources are for all kinds of industrial purposes [41]. Therefore, the five types are integrated into industrial product supply. Second, this paper deletes seven types of land resources carriers, including genetic resources supply, ornamental resources supply, social insurance, employment guarantee, air quality regulation, water regulation, moderation of extreme events, and spiritual and historic carriers because it is difficult to find out their corresponding human demands. Third, because four types of land resources carriers including soil retention, nutrient cycling, primary productivity, and pollination do not directly serve human activities [49], these are therefore deleted. Fourth, scientific and educational carrier is integrated with cultural and artistic carrier, and becomes educational and literary carrier; and leisure carrier is integrated with aesthetic carrier, becoming leisure and aesthetic carrier. As a result, the 11 types of land resources carriers left under the three primary functions are shown in Table 2.

Land resources carriers are considered to be able to satisfy the main human demands, and therefore they are used to describe LRCC.

### 2.2. Land Resources Loads

Land resources loads will be identified by analyzing the carrying relationships between human demands and land use functions. In fact, humans are fully dependent on the land resources system and functions that it provides, such as food, freshwater, aesthetic enjoyment, and climate regulation [43]. Therefore, food supply, freshwater supply, climate regulation, and leisure and aesthetic carrier correspond to various carrying loads including food demand, freshwater demand, demand for carbon emission, and leisure and aesthetic demand, respectively. Similarly, the land resources loads corresponding to the other seven remaining carriers can be defined. As a result, land resources loads are formulated and presented in Table 3.

## 3. LRCC Evaluation Model

This section will introduce a set of measurement models for evaluating sub-LRCCs and LRCC. In order to develop these evaluation models, indicators for measuring land resources carriers and indicators for measuring land resources loads will be selected based on production function, living function, and ecological function perspectives.

### 3.1. Indicators for Measuring Land Resources Carriers

By referring to land resources carriers shown in Table 2, this section will select an indicator against each type of land resources carrier. These indicators can be measured in different spatial scales (e.g., scales of land parcel, grid, or administrative region and so on) and temporal scales (e.g., second, minute, hour, or year and so on).

#### 3.1.1. Land Resources Carrier Indicators from Production Function Perspective

There are four types of land resources carriers from a production function perspective, as shown in Table 2, including food supply, freshwater supply, industrial product supply, and commercial services supply. For food supply, it includes production of fish, game, grains, nuts, and fruits [40]. According to the report by Chinese Nutrition Society [50], food supply was divided into four types including supplies of grains, animal foods, processed foods, as well as vegetables and fruits. Based on the previous study on assessing for multiple land use functions, trade-offs and synergies among multiple land use functions, and land use production-living-ecological functions, the function of food supply can be measured by a single indicator or multiple indicators. Single indicators can be per unit area grain yield [51], or per capita total agricultural output values [52]; and multiple indicators can be per unit area grain yield, per unit area timber yield, and per unit area aquatic product yield [53]. Therefore, this paper selects total grain yield as the indicator to measure the function of food supply in administrative region j at year t, and this indicator is expressed by $C_{P,1}$ (unit: kg).

For a freshwater supply coded by $C_{P,2}$, it actually means available freshwater supply, because not all freshwater resources cannot be fully utilized by humans. Available freshwater is the sum of available surface freshwater and underground exploitable freshwater. This paper calculates the volume of available freshwater resources based on the volume of freshwater resources through the principle shown in the study of Ministry of Water Resources of the People’s Republic of China [54], which can be expressed as follows:(1)$$C_{P,2}=W_{1}+W_{2}$$
(2)$$W_{1}=Vol\times k\times \lambda$$
(3)$$W_{2}=Vol\times (1-k)\times \beta$$
where $C_{P,2}$ is the volume of available freshwater resources in administrative region j at year t (unit: m^3^·yr^−1^); $W_{1}$ denotes available surface freshwater resources in administrative region j at year t (unit: m^3^·yr^−1^); $W_{2}$ represents underground exploitable freshwater resources in administrative region j at year t (unit: m^3^·yr^−1^); Vol is the volume of freshwater resources in administrative region j at year t (unit: m^3^·yr^−1^); k denotes the proportion of surface freshwater resources (unit: %); λ is the availability rate of surface freshwater resources (unit: %); β represents the exploitability rate of underground freshwater resources (unit: %).

In referring to industrial product supply, it includes the processing products of secondary industry, such as shoes, clothes, and furniture. Zou et al. [55] used the gross output value of secondary industry to measure industrial product supply. However, it is difficult to find out per capita gross output value of secondary industry as the human demand for industrial product supply. As the products of secondary industry are processed in various industrial plants built on land, this paper selects land area which supports various industrial plants as the indicator coded by $C_{P,3}$ (unit: m^2^) to measure the function of industrial product supply in administrative region j at year t.

For commercial services supply, it means providing retail, catering, and other types of services. Fan et al. [53] used commercial service point density to measure the function of commercial services supply. Zou et al. [55] used the total output values of the wholesale and retail industry, accommodation and catering industry, as well as real estate industry to measure this function of commercial services supply. Ma et al. [56] used the land area of commercial services to measure this function. Therefore, land area supporting commercial services is selected to measure the function of commercial services supply in administrative region j at year t in this paper, and the indicator is expressed by $C_{P,4}$ (unit: m^2^).

#### 3.1.2. Carrier Indicators from Living Function Perspective

There are five types of land resources carriers to support the living function perspective, as shown in Table 2, including housing supply, transportation services supply, supply of public administration and public services, educational and literary carriers, as well as leisure and aesthetic carriers. For housing supply, it means providing the place of life for urban and rural residents. In studying a rural region, Yang et al. [57] integrated multi-factors, including rural per capita net income, rural power facility, and rural residential land area to measure housing supply in a rural region. In line with this reference, this paper selects residential land area to measure housing supply in administrative region j at year t, and this indicator is expressed as $C_{L,1}$ (unit: m^2^).

For transportation services supply, it refers to freight services and passenger services provided by bus, car, taxi, light rail and other types of transportation means carried by expressway, trunk road, secondary trunk road, branch, land for light rail, land for passenger station, land for parking lot, and other types of transportation lands. Zhou et al. [52] used the transportation land area to measure the function of transportation services supply. Land area for transportation services coded by $C_{L,2}$ (unit: m^2^) is selected in this paper to measure transportation services supply in administrative region j at year t.

For the supply of public administration and public services, it refers to three types of land use, including land for organ groups, the press and publishing, and public facilities, which can provide many functions, such as the availability of water supply and water sewer. Ma et al. [56] used land area for public administration and public services to measure this supply. Therefore, land area for public administration and public services is selected to measure the supply of public administration and public services in administrative region j at year t in this paper, and the indicator is expressed by $C_{L,3}$ (unit: m^2^).

For educational and literary carriers, it refers to land resources for schools, scientific research, libraries, and exhibition halls. Therefore, the land area of science, education, and culture is selected to measure educational and literary carriers in administrative region j at year t in this paper, and the indicator is expressed by $C_{L,4}$ (unit: m^2^).

For leisure and aesthetic carriers, it refers to green spaces which can accommodate walking sports and the demand for the enjoyment of scenery. Thus, green space can play a role in maintaining mental and physical health [44]. Green spaces contain all urban green, agricultural green, forests and nature conservation areas [58]. The land area of urban green and agricultural green is therefore selected to measure leisure and aesthetic carriers in administrative region j at year t in this paper, and the indicator is expressed by $C_{L,5}$ (unit: m^2^). Specifically, this land area includes the park, zoo, botanical garden, square, and so on.

#### 3.1.3. Carrier Indicators from an Ecological Function Perspective

There are two types of land resources carriers from the ecological function perspective, as shown in Table 2, including climate regulation and waste purification. For climate regulation, it means the maintenance of a favorable climate (e.g., temperature, precipitation, etc.) for human habitation, health, and cultivation [47]. Previous studies usually applied carbon storage as an indicator to measure climate regulation [53,59,60]. The calculation model for the value of this indicator is presented as follows:(4)$$C_{E,1}=\sum_{1}^{n}C\_total_{xyz}$$
where $C_{E,1}$ is total carbon storage in administrative region j at year t (unit: ton·yr^−1^); *n* denotes total number of grids in an administrative region; $C\_total_{xyz}$ represents total carbon storage in grid (*x*, *y*) with land use type *z* at year t (unit: ton·yr^−1^), it can be calculated by using InVEST model which is explained by *Natural Capital Project* (http://releases.naturalcapitalproject.org/invest-userguide/latest/carbonstorage.html, accessed on 4 October 2021).

For waste purification, it means that land resources such as microorganisms in the soil can break down most waste through their biological activity [44]. This paper selects the volume of nutrients’ retention, including retention of total nitrogen (TN) and total phosphorus (TP), to measure the function of waste purification. The calculation for the value of this indicator is from our own Addition and Subtraction Method according to $X_{\exp,xyz}$. The equations are shown as follows:(5)$$C_{E,2}=\sum_{1}^{n}X_{ret,xyz}$$
(6)$$X_{ret,xyz}=load_{xyz}\times A_{xy}-X_{\exp,xyz}$$
where $C_{E,2}$ is the total volume of TN retention or total volume of TP retention in administrative region j at year t (unit: kg·yr^−1^); n denotes the total number of grids in an administrative region; $X_{ret,xyz}$ represents total volume of TN retention or total volume of TP retention in grid (*x*, *y*) with land use type *z* at year t (unit: kg·ha^−1^·yr^−1^); $load_{xyz}$ is the total volume of TN or TP load in grid (*x*, *y*) with land use type *z* at year t (unit: kg·ha^−1^·yr^−1^), which can be collected by referring to previous studies; $A_{xy}$ denotes the area of the grid (*x*, *y*) which is the same (unit: ha^−1^); $X_{\exp,xyz}$ represents the total volume of TN export or total volume of TP export in administrative region *j* at year *t* (unit: kg·yr^−1^), it is explained by *Natural Capital Project* (http://releases.naturalcapitalproject.org/invest-userguide/latest/ndr.html, accessed on 4 October 2021).

From the carrier indicators mentioned above, there is a summary in Table 4 as follows.

### 3.2. Indicators for Measuring Land Resources Loads

By referring to the land resources loads specified in Table 3, this section will examine and select the indicator of land resources loads against each type of land resources function. These indicators are measured in different spatial and temporal scales, spatially at administrative region in a year.

#### 3.2.1. Land Resources Load Indicators from Production Function Perspective

There are four types of land resources loads supported by the production function, as shown in Table 3, including food demand, freshwater demand, industrial product demand, and commercial services demand. For food demand, it can be measured by the indicator of per capita grain consumption in administrative region j at year t, coded as $L_{P,1}$ (unit: kg/person). This indicator can be divided into per capita direct grain consumption and per capita indirect grain consumption [61]. Per capita direct grain consumption refers to per capita grain consumption for eating, coded as $F_{1}$. Per capita indirect grain consumption includes four elements: (1) per capita grain consumption for producing industrial products (e.g., liquor and beer), coded as $F_{2}$; (2) per capita grain consumption for breeding hogs, poultry, fishery animals and other types of animals, expressed as $F_{3}$; (3) per capita grain consumption for sowing, such as middle rice, soybean, and corn, coded by $F_{4}$; (4) and per capita grain consumption for loss during storage and transport process, expressed as $F_{5}$. Therefore, the calculation for the value of the indicator $L_{P,1}$ can be obtained by using the models introduced by Tang and Li [61].

For freshwater demand, it can be considered as per capita freshwater consumption in administrative region j at year t, which is expressed by $L_{P,2}$ (unit: m^3^/person). The value of this indicator can be calculated by using the volume of freshwater consumption ($W_{D}$) divided by the total number of the permanent population ($P_{R}$). 

In referring to the load “industrial product demand”, its value in administrative region j at year t can be measured by using the indicator of demand for industrial land area per capita, which is coded by $L_{P,3}$ (unit: m^2^/person). This indicator can refer to the planning land area per capita.

For commercial services demand, its value is measured by the indicator of demand for commercial land area per capita in administrative region j at year t, denoted by $L_{P,4}$ (unit: m^2^/person). Similarly, $L_{P,4}$ can refer to the planning land area per capita.

#### 3.2.2. Load Indicators from Living Function Perspective

There are five types of land resources loads supported by land resources living function, as shown in Table 3, including housing demand, transportation services demand, demand for public administration and public services, educational and literary demand, as well as leisure and aesthetic demand. For housing demand in administrative region j at year t, it can be measured by the indicator of demand for residential land area per capita, coded by $L_{L,1}$ (unit: m^2^/person). As the floor area ratio in urban residential land is higher than that in rural region, $L_{L,1}$ is divided into two types: $A_{R}$ and $A_{U}$, they represent demand for rural residential land area per capita and demand for urban residential land area respectively. The value of $A_{U}$ can be obtained by referring to planning land area per capita, and the value of $A_{R}$ can be calculated by the ratio index of rural residential land area and rural permanent population.

In referring to transportation services demand, it refers to demand for transportation land area per capita in administrative region j at year t, coded as $L_{L,2}$ (unit: m^2^/person). This indicator can refer to the planning land area per capita.

In referring to the demand for public administration and public services, its value will be measured by the indicator of per capita demand for a land area supporting public administration and public services in administrative region j at year t, which is expressed by $L_{L,3}$ (unit: m^2^/person). Similarly, $L_{L,3}$ can refer to the planning land area per capita.

In referring to educational and literary demand, its value will be measured by the indicator of per capita demand for a land area carrying science in administrative region j at year t, which is expressed by $L_{L,4}$ (unit: m^2^/person). Similarly, $L_{L,4}$ can refer to the planning land area per capita.

For leisure and aesthetic demand, its value will be measured by the indicator of per capita demand for a land area supporting urban green space and agricultural green space in administrative region j at year t, which is coded as $L_{L,5}$ (unit: m^2^/person). Similarly, $L_{L,5}$ can refer to the planning land area per capita.

#### 3.2.3. Load Indicators from Ecological Function Perspective

There are two types of land resources loads carried by land resources ecological function, as shown in Table 3, including carbon emission and waste discharge. In referring to carbon emission, it is measured by the indicator of per capita carbon emission in administrative region j at year t, coded by $L_{E,1}$ (unit: ton/person). The value of this indicator can be obtained by carbon emission volume ($T_{C}$) dividing total number of permanent population ($P_{R}$).

For waste discharge, its value is measured by the indicator of per capita waste discharge, which is expressed by $L_{E,2}$ (unit: kg/person). The calculation for the value of this indicator is presented as follows:(7)$$L_{E,2}=\frac{\sum_{1}^{n}load_{xyz}\times A_{xy}}{P_{R}}$$
where $load_{xyz}$ is the total volume of waste discharge (TN or TP load) in grid (*x*, *y*) with land use type *z* at year t (unit: kg·ha^−1^·yr^−1^), which can be collected by referring to previous studies; n denotes total number of grids in an administrative region; $A_{xy}$ denotes the area of the grid (*x*, *y*) which is the same (unit: ha^−1^).

From the load indicators mentioned above, there is a summary in Table 5 as follows.

### 3.3. Measurement Models for Land Resources Carrying Capacity

According to the definition of LRCC given in Section 2, LRCC refers to the maximum number of populations that can be carried by land resources without the transformation of the land resources system. Table 2 and Table 3 show that there are eleven types of land resources carriers and loads. Before establishing the general measurement model for land resources carrying capacity, it is necessary to construct the measurement models for measuring the capacity at a sub-functional level based on relationships between individual land resources carriers and loads, as shown as follows:(8)$$P_{P,i}=\frac{C_{p,i}}{L_{p,i}}$$
(9)$$P_{L,i}=\frac{C_{L,i}}{L_{L,i}}$$
(10)$$P_{E,i}=\frac{C_{E,i}}{L_{E,i}}$$
where $P_{P,i}$, $P_{L,i}$, $P_{E,i}$ is the maximum number of populations that can be carried by land resources carrier i from production, living, and ecological function perspective in administrative region j at year t.

Consequently, the capacity at the primary functional level is obtained as follows:(11)$$P_{P}=\sum_{i=1}^{n_{P}}P_{P,i}W_{P,i}$$
(12)$$P_{L}=\sum_{i=1}^{n_{L}}P_{L,i}W_{L,i}$$
(13)$$P_{E}=\sum_{i=1}^{n_{E}}P_{E,i}W_{E,i}$$
where $W_{P,i}$, $W_{L,i}$, and $W_{E,i}$ are the weighting value of ratio index $P_{P,i}$, $P_{L,i}$, and $P_{E,i}$ respectively, which can be calculated by referring to the analytic hierarchy process (AHP) proposed by Saaty [62].

Finally, LRCC can be obtained as follows:(14)$$P=P_{P}W_{P}+P_{L}W_{L}+P_{E}W_{E}$$
where P is the maximum number of populations can be carried by all types of land resources carriers in administrative region j at year t, namely LRCC; $W_{P}$, $W_{L}$, and $W_{E}$ are the weighting value of ratio index $P_{P}$, $P_{L}$, and $P_{E}$ respectively, which can be obtained through AHP.

In using the AHP method, a complex problem will be structured with several layers, e.g., the objective layer, category layer, and indicator layer. In this study, there are three layers, including objective layer, category layer, and indicator layer. The LRCC (coded by P) is regarded as the objective layer. The category layer includes $P_{P}$, $P_{L}$, and $P_{E}$, as well as the indicator layer contains $P_{P,i}$, $P_{L,i}$, and $P_{E,i}$. Numbers ranging from 1-9 and their reciprocals are used to indicate the relative importance of these measurement elements for LRCC in the pairwise comparison. Number 1, 3, 5, 7, and 9 denote equal importance, weak importance, essential importance, demonstrated importance, and absolute importance of one over another, respectively [62]. Numbers 2, 4, 6, 8 are the mean intermediate values between the two adjacent judgments. A judgment matrix is constructed by the degrees of relative importance of these measurement elements. Finally, the judgment matrix is calculated by mathematical theory, resulting in the vector relating to the largest eigenvalue of the judgment matrix. If the consistency ratio (CR) of the judgment matrix is less than 0.1, the calculated vector is the weights of these measurement elements. 

## 4. Case Demonstration

In this section, a demonstration in the Chinese context is used to show the application and effectiveness of land resources carrying capacity (LRCC) evaluation model established in Equations (1)–(14).

### 4.1. Study Area

This study is conducted in the main city area of Chongqing (the municipality of China), which is located in the western portion of Southwest China (see Figure 2). The main city area includes nine county-level administration districts, namely Yuzhong, Dadukou, Jiangbei, Shapingba, Jiulongpo, Nan’an, Beibei, Yubei, and Ba’nan districts. The main city area in Chongqing is a place where there are four parallel north-south mountains, namely the Jinyun, Zhongliang, Tongluo, and Mingyue mountains, as well as two west-east rivers, namely the Jialing and Yangtze Rivers. The nine county-level administration districts are experiencing the fastest processes of economic growth and migration in Chongqing. GDP in the main city area reached to 933.42 billion yuan in 2019, with a permanent population of 8,843,900 [63]. The economy has traditionally relied on industry, and the commercial and tourism industries are also well developed. The main city area progressively evolved into a polycentric city composed of “one central urban, six subcenters, and twenty-one urban clusters”.

### 4.2. Data

The data used in this paper are divided into geographic and socioeconomic data. Specifically, the data type, source, time point, and resolution are shown in Table 6.

In order to obtain the value for all indicators, these data in Table 6 need to be processed, and the procedures for data processing are also shown in the Appendix A.

### 4.3. Results

By applying the data presented in Appendix A to the measurement models of Equations (8)–(10), the values of $P_{P,i}$, $P_{L,i}$, and $P_{E,i}$ can be obtained, as shown in Table 7.

To obtain the value of $P_{P}$, $P_{L}$, $P_{E}$, and P, the values of $W_{P,i}$, $W_{L,i}$, $W_{E,i}$, $W_{P}$, $W_{L}$, and $W_{E}$ are determined by using the AHP method. In referring to Chongqing Urban and Rural Master Plan During 2007–2020, it can be seen that the plan of the main city area is to develop a modern service industry (e.g., software R&D, warehousing and distribution), as well as a modern manufacturing industry (e.g., electronic information and high-end equipment). On the other hand, the plan is to build large residential areas and improve supporting facilities for residents. Therefore, the judgment matrixes of pairwise in the indicator layer and sub-indicator layer are shown in Table 8, Table 9, Table 10 and Table 11, respectively.

In using AHP method, it is necessary to check the CR, the CR of the judgment matrixes shown in Table 8, Table 9, Table 10 and Table 11 is 0 respectively (less than 0.1). Therefore, the value of $W_{P}$, $W_{L}$, and $W_{E}$ is 0.455, 0.455, and 0.09 respectively, and the value of $W_{P,1}$, $W_{P,2}$, $W_{P,3}$, and $W_{P,4}$ is 0.046, 0.318, 0.318, and 0.318 respectively. The elements of $W_{L,i}$ carry same value, namely 0.2, and it is the same to $W_{E,i}$, valuing 0.5. By applying these weighting values and individual LRCCs shown in Table 7 into Equations (11)–(14), the value of $P_{P}$, $P_{L}$, $P_{E}$, and P can be obtained, as presented in Table 12.

## 5. Discussion

The land resources carrying capacity (LRCC) is not only related to the supply capacity provided by land resources (namely land resources carriers), but also related to the pressure generated by human activities (namely land resources loads). Therefore, this study introduces a “load-carrier” perspective method for evaluating LRCC through examining the relationships between land resources carriers and loads. The theory of multifunctional land use has been used as the theoretical base to determine land resources carriers, loads, and their measurement indicators. As a tool for examining sustainable land resources use, a “load-carrier” perspective evaluation method can verify what number of current populations are supported by land resources, and identify and address the potential overloaded risk in order to keep land resources demands imposed by human activities under control [76].

The successful application of the introduced “load-carrier” perspective method requests the proper understanding of land resources carriers and land resources loads. When an individual land resources carrier is referred, its corresponding load should be properly assigned. In other words, the carrier can provide the materials or services that human beings need. Secondly, the indicators for measuring land resources carriers and loads must be selected properly. Furthermore, the weighting values between three dimensional capacities (production dimension $P_{P}$, living dimension $P_{L}$, and ecological dimension $P_{E}$) will be different when different study areas are referred, as social-economic backgrounds and development plans are different. For example, in referring to the case demonstration, the main city area focuses on the developing production and living functions of land resources, and some others in Chongqing focus on developing the ecological function of land resources. Therefore, when the weighting values of $P_{P}$, $P_{L}$, and $P_{E}$ in the main city area are considered, $P_{P}$ and $P_{L}$ are of essential importance to $P_{E}$.

The proposed “load-carrier” perspective method is proved effective through the case demonstration, which suggests that the LRCC in Yuzhong, Dadukou, Jiangbei, Shapingba, Jiulongpo, Nan’an, Beibei, Yubei, and Ba’nan is 14.686, 20.855, 50.505, 77.101, 76.4, 58.692, 59.525, 171.7, and 111.836 (unit: 10,000 person), respectively, in 2019 (see Table 12). When compared with the population in 2019 (66.24, 36.2, 90.28, 116.5, 123.3, 92.8, 81.6, 168.35, and 109.12, respectively), according to the report by the Chongqing Statistical Bureau [63], it reveals that seven regions are overloaded in the LRCC, and the other two regions are approaching the limitation of LRCC. The causes for the poor LRCC in the surveyed regions can be further explored by analyzing their capacity at the sub-functional dimension, namely, $P_{P,i}$, $P_{L,i}$, and $P_{E,i}$. For example, Yuzhong is identified as an overloaded region with the P value of 14.686 compared with the value of 66.24 (unit: 10,000 person). By further exploring Yuzhong’s capacity at the sub-functional dimension, it is considered that all the values of the capacities at the sub-functional dimension are less than the value of 66.24. The situation indicates that Yuzhong district cannot support the existing population under the level of per capita demand and the supply capacity of land resources. On the other hand, the “advantage” and “disadvantage” of LRCC can be revealed in the case demonstration. For example, in referring to 2019, the “advantage” of LRCC in Yuzhong is the capacity at the function of waste purification ($P_{E,2}$) and the “disadvantage” is the capacity at the function of food supply ($P_{P,1}$) (see Table 7). This finding is consistent with the actual situation. As Yuzhong County has been fully urbanized, there is no cultivated land. Therefore, the volume of nutrients export (TN and TP export) caused by agriculture is small, and no grain is produced.

It is therefore suggested that the local governments in the case study regions should consider introducing policy measures from a carriers or loads perspective. For example, when facing the shortage of the capacities at the functions of food supply, climate regulation, or waste purification, the local governments in the case study regions such as Yuzhong can introduce policy measures from the carriers’ perspective. It should increase grain import trade and increase environmental investment in terms of governing carbon emission, TN and TP discharge. On the other hand, when facing the shortage of the capacities in various functions, including industrial product supply, commercial services supply, housing supply, educational and literary carriers, or leisure and aesthetic carriers, the local governments in the case study regions such as Yuzhong can introduce policy measures from the loads’ perspective. It should advocate that inhabitants decrease land resources demands or obtain demands from other regions, such as building high-rise apartments, office skyscrapers, or smaller-size accommodations. Actually, as reported by Liu et al. [77], Yuzhong has been implementing the scheme of massive demolition and the rebuilding of structures, resulting in high-rise apartments and office skyscrapers, and inhabitants face the challenge of dense buildings and severely congested traffic.

## 6. Conclusions

Proper evaluations on LRCC requests for the consideration of both carriers and loads. The “load-carrier” perspective method introduced in this paper can meet this requirement. 

Secondly, the composition of land resources carriers and loads embodied in the load-carrier perspective method is determined by using the theory of multifunctional land use. For other regions of China, the proposed indicators don’t need to be modified. Thirdly, the demonstration case has further shown the effectiveness of using this method to obtain the value of LRCC. The case results suggest that seven regions are overloaded in LRCC and the other two regions are approaching the limitation of LRCC among 9 county-level administration regions in Chongqing.

The theoretical value of this study is the contribution to the development of literature in the field of land resources carrying capacity. The “load-carrier” perspective adopted in developing the measurement method provides new insights on both carriers and loads in a land resources system. Practically, the research results provide important references to the local governments in the case study regions for understanding the number of populations that can be carried by their land resources at a specific time point, thus tailor-made policies can be formulated to ensure that land resources demands imposed by human activities are under control at a reasonable level.

There are some limitations to this study. Data collection and processing for the case study involves very complicated calculation procedures, thus the accuracy of calculation results may be affected. It is recommended that the process of data collection and processing should be improved in the future.

## Figures and Tables

**Figure 1 ijerph-19-05503-f001:**
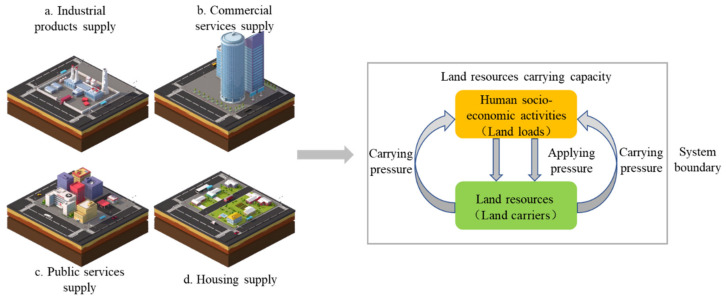
The concept of land resources carrying capacity.

**Figure 2 ijerph-19-05503-f002:**
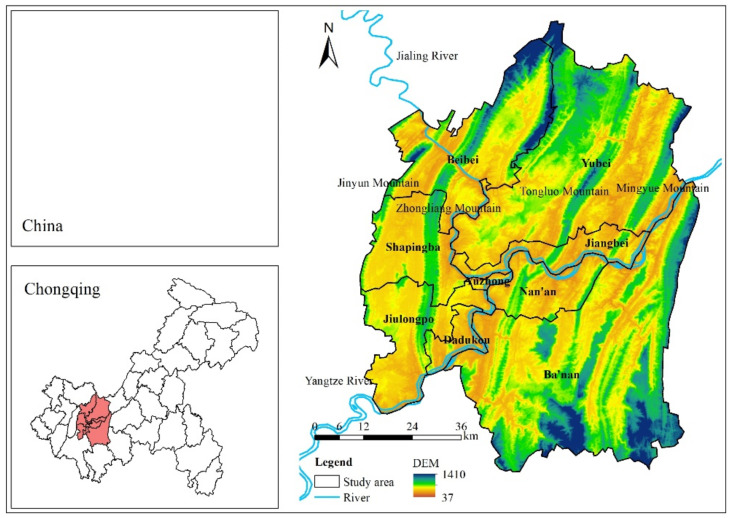
Study area.

**Table 1 ijerph-19-05503-t001:** The comprehensive classification of land resources carriers.

Primary Functions	Sub-Functions
Production function	Food supply
	Freshwater supply
	Medicinal resources supply
	Genetic resources supply
	Timber production
	Fiber production
	Ornamental resources supply
	Provision of energy resources
	Minerals production
	Industrial product supply
	Commercial services supply
Living function	Housing supply
	Transportation services supply
	Supply of public administration and public services
	Social insurance
	Employment guarantee
	Carrier function of Science and education
	Leisure carrier
	Carrier function of culture and artist
	Aesthetic carrier
	Spiritual and historic carrier
Ecological function	Air quality regulation
	Climate regulation
	Water regulation
	Waste purification
	Moderation of extreme events
	Pollination
	Soil retention
	Nutrient cycling
	Primary productivity

Source: Adapted from Li and Fang [39].

**Table 2 ijerph-19-05503-t002:** Land resources carriers.

Primary Functions	Sub-Functions
Production function	Food supply
	Freshwater supply
	Industrial product supply
	Commercial services supply
Living function	Housing supply
	Transportation services supply
	Supply of public administration and public services
	Educational and literary carrier
	Leisure and aesthetic carrier
Ecological function	Climate regulation
	Waste purification

**Table 3 ijerph-19-05503-t003:** Land resources loads.

Primary Functions	Sub-Functions	Land Resources Loads
Production function	Food supply	Food demand
	Freshwater supply	Freshwater demand
	Industrial product supply	Industrial product demand
	Commercial services supply	Commercial services demand
Living function	Housing supply	Housing demand
	Transportation services supply	Transportation services demand
	Supply of public administration and public services	Demand for public administration and public services
	Educational and literary carrier	Educational and literary demand
	Leisure and aesthetic carrier	Leisure and aesthetic demand
Ecological function	Climate regulation	Carbon emission
	Waste purification	Waste discharge

**Table 4 ijerph-19-05503-t004:** Indicators for measuring land resources carriers.

Primary Functions	Sub-Functions	Indicators
Production function	Food supply	Total grain yield ($C_{P,1}$)
	Freshwater supply	Available freshwater supply ($C_{P,2}$)
	Industrial product supply	Land area which supports various industrial plants ($C_{P,3}$)
	Commercial services supply	Land area supporting commercial services ($C_{P,4}$)
Living function	Housing supply	Residential land area ($C_{L,1}$)
	Transportation services supply	Land area for transportation services ($C_{L,2}$)
	Supply of public administration and public services	Land area for public administration and public services ($C_{L,3}$)
	Educational and literary carrier	Land area of science, education, and culture ($C_{L,4}$)
	Leisure and aesthetic carrier	Land area of urban green and agricultural green ($C_{L,5}$)
Ecological function	Climate regulation	Carbon storage ($C_{E,1}$)
	Waste purification	The volume of nutrients’ retention ($C_{E,2}$)

**Table 5 ijerph-19-05503-t005:** Indicators for measuring land resources loads.

Primary Functions	Sub-Functions	Indicators
Production function	Food demand	Per capita grain consumption ($L_{P,1}$)
	Freshwater demand	Per capita freshwater consumption ($L_{P,2}$)
	Industrial product demand	Demand for industrial land area per capita ($L_{P,3}$)
	Commercial services demand	Demand for commercial land area per capita ($L_{P,4}$)
Living function	Housing demand	Residential land area per capita ($L_{L,1}$)
	Transportation services demand	Demand for transportation land area per capita ($L_{L,2}$)
	Demand for public administration and public services	Per capita demand for land area supporting public administration and public services ($L_{L,3}$)
	Educational and literary demand	Per capita demand for land area carrying science ($L_{L,4}$)
	Leisure and aesthetic demand	Per capita demand for land area supporting urban green space and agricultural green space ($L_{L,5}$)
Ecological function	Carbon emission	Per capita carbon emission ($L_{E,1}$)
	Waste discharge	Per capita waste discharge ($L_{E,2}$)

**Table 6 ijerph-19-05503-t006:** Data description.

Data Type	Data Source	Time Point	Resolution
**Geographic data**			
The third Chongqing land use survey database	Chongqing Bureau of Natural Resources and Planning	2019	Vector
Carbon density	References [64,65,66,67,68,69,70,71,72,73]		County level
DEM (ASTER GDEM V2)	Geospatial Data Cloud (http://www.gscloud.cn/, accessed on 4 October 2021)	2019	30 m × 30 m
The boundary of main urban area	Resource and Environment Science Data Center, Chinese Academy of Sciences	2015	Vector
Precipitation	Meteorological data center of China Meteorological Administration	2019	Vector: Site
NDR: Biophysical_table, including: (1) the total volume of waste discharge in grid (*x*, *y*) with land use type *z* (*load_xyz_*); (2) maximum retention efficiency in grid (*x*, *y*) with land use type *z*; and (3) critical length in grid (*x*, *y*) with land use type *z*)	Reference [74]	2019	County level
(1) Threshold flow accumulation; (2) Borselli k parameter; (3) subsurface critical length; and (4) Subsurface maximum retention efficiency	InVEST User’s Guide (https://naturalcapitalproject.stanford.edu/software/invest, accessed on 4 October 2021)	2019	County level
**Socioeconomic data**			
(1) Total grain yield (*C_P_*_,1_); (2) Total number of permanent population (*P_R_*); and (3) Total number of rural permanent population	Chongqing Statistical Yearbook	2019	County level
(1) The proportion of urban or rural permanent population; (2) Per capita grain household consumption for eating in urban or rural area; (3) Total output of liquor, beer, poultry eggs, aquatic products, pork, or poultry meats; (4) Number of hogs; (5) The sown area of middle rice, soybean, or corn	Chongqing Statistical Yearbook	2019	Municipality level
The total seed volume of middle rice, soybean, or corn per sown area	Cost-Benefit Compilation of Chinese Agricultural Products	2005–2018	Municipality level
The proportion of grain non-household consumption in grain household consumption for urban or rural area	Reference [75]		Municipality level
(1) The grain consumption coefficient of liquor, beer, poultry eggs, aquatic products, pork, or poultry meats; (2) The grain consumption per hog; (3) The loss rate during storage and transport process	Reference [61]		
(1) The volume of freshwater resources (*Vol*); (2) The volume of freshwater resources consumption (*W_D_*); and (3) The proportion of surface freshwater resources (*k*)	Chongqing Water Resources Bulletin	2019	County level
(1) The availability rate of surface freshwater resources (*λ*) and (2) The exploitability rate of underground freshwater resources (*β*)	A Guide to Water Resources Assessment: SL/T 238-1999	1999	County level
(1) Demand for industrial land area per capita (*L_P_*_,3_); (2) Demand for commercial land area per capita (*L_P_*_,4_); (3) Demand for transportation land area per capita (*L_L_*_,2_); (4) Per capita demand for land area supporting public administration and public services (*L_L_*_,3_); (5) Per capita demand for land area carrying science, education, and culture (*L_L_*_,4_); and (6) Per capita demand for land area supporting urban green space and agricultural green space (*L_L_*_,5_)	Chongqing Urban and Rural Master Plan During 2007–2020	2020	The main city area
Demand for urban residential land area per capita (*A_U_*)	Code for Classification of Urban Land Use and Planning Standards of Development Land: GB 50137-2011		The main city area
Carbon emission volume (*T_C_*)	Carbon Emission Accounts and Datasets (https://www.ceads.net.cn/, accessed on 4 October 2021)	1997–2017	County level

**Table 7 ijerph-19-05503-t007:** The value of $P_{P,i}$, $P_{L,i}$, and $P_{E,i}$ in the nine county-level administration districts in 2019 (unit: 10,000 people).

	$P_{P,1}$	$P_{P,2}$	$P_{P,3}$	$P_{P,4}$	$P_{L,1}$	$P_{L,2}$	$P_{L,3}$	$P_{L,4}$	$P_{L,5}$	$P_{E,1}$	$P_{E,2}$
Yuzhong	0.0000	4.6430	0.0410	18.8351	20.5434	32.5444	13.3195	18.6701	22.6780	1.1452	31.6320
Dadukou	0.2025	11.8052	11.0271	15.2404	39.7904	65.6526	24.8910	9.9767	16.6825	8.5767	14.8014
Jiangbei	0.5830	25.0359	17.5694	74.6849	88.9139	107.6017	73.5709	30.2685	36.9834	22.8590	39.6764
Shapingba	2.4366	71.9453	22.8635	98.2065	107.8301	140.0911	70.7898	142.9966	25.3714	36.2591	69.9993
Jiulongpo	2.9236	70.1171	35.4789	119.3752	115.5802	133.0377	71.2099	55.2688	36.7610	47.9511	91.8119
Nan’an	0.4899	34.8077	14.4781	69.3738	113.3515	126.6463	72.8655	52.3252	44.7247	27.3563	65.9751
Beibei	10.5490	67.5863	26.9236	43.0151	95.2710	103.5771	67.5331	52.6005	51.5197	81.1373	45.0840
Yubei	26.0747	225.4788	44.2634	151.7002	239.7952	405.9373	241.0800	92.6837	101.5974	151.2670	110.1589
Ba’nan	51.0753	262.4225	24.6275	105.7128	106.0398	113.7958	68.6010	50.8582	37.8689	365.0663	70.3207

Note: $P_{E,2}$ is the average value between the total populations carried by TN waste purification and total populations supported by TP waste purification.

**Table 8 ijerph-19-05503-t008:** Judgment matrix in the category layer.

	$P_{P}$	$P_{L}$	$P_{E}$
$P_{P}$	1	1	5
$P_{L}$	1	1	5
$P_{E}$	1/5	1/5	1

**Table 9 ijerph-19-05503-t009:** Judgment matrix in the indicator layer ($P_{P,1}$ –$P_{P,4}$ ).

	$P_{P,1}$	$P_{P,2}$	$P_{P,3}$	$P_{P,4}$
$P_{P,1}$	1	1/7	1/7	1/7
$P_{P,2}$	7	1	1	1
$P_{P,3}$	7	1	1	1
$P_{P,4}$	7	1	1	1

**Table 10 ijerph-19-05503-t010:** Judgment matrix in the indicator layer ($P_{L,1}$ –$P_{L,5}$ ).

	$P_{L,1}$	$P_{L,2}$	$P_{L,3}$	$P_{L,4}$	$P_{L,5}$
$P_{L,1}$	1	1	1	1	1
$P_{L,2}$	1	1	1	1	1
$P_{L,3}$	1	1	1	1	1
$P_{L,4}$	1	1	1	1	1
$P_{L,5}$	1	1	1	1	1

**Table 11 ijerph-19-05503-t011:** Judgment matrix in the indicator layer ($P_{E,1}$ –$P_{E,2}$ ).

	$P_{E,1}$	$P_{E,2}$
$P_{E,1}$	1	1
$P_{E,2}$	1	1

**Table 12 ijerph-19-05503-t012:** The value of LRCC in the 9 county-level administration districts in 2019. (unit: 10,000 person).

	$P_{P}$	$P_{L}$	$P_{E}$	P
Yuzhong	7.484	21.551	16.389	14.686
Dadukou	12.124	31.399	11.689	20.855
Jiangbei	37.348	67.468	31.268	50.505
Shapingba	61.528	97.416	53.129	77.101
Jiulongpo	71.719	82.372	69.881	76.400
Nan’an	37.780	81.983	46.666	58.692
Beibei	44.240	74.100	63.111	59.525
Yubei	135.289	216.219	130.713	171.700
Ba’nan	127.301	75.433	217.694	111.836

## Data Availability

The data that support the findings of this study are available from the corresponding author, upon reasonable request.

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
