# Peer review of "A Load-Carrier Perspective Method for Evaluating Land Resources Carrying Capacity"

_ijerph, 2022, doi:10.3390/ijerph19095503_

Round 1

Reviewer 1 Report

The study presents a new evaluation method from a load-carrier perspective for assessing land resources carrying capacity (LRCC) by examining the relationships between the pressure caused by human activities and the supply capacity of land resources. In developing this method, a land resources system is determined by two primary components, carrier and load. The results of the study are shown on a case study including 9 county-level administration regions. The case results shows that seven regions are overloaded in LRCC and the other two regions are approaching to the limitation of LRCC.

The method and results are generally well explained. AHP should be better explained to those that are not familiar with this method. Values obtained in Table 5 and Table 10 and their interpretation i.e. the meaning should be better explained near the results.

There are some typos such as line 33:  industrial activities; On the other hand.

Line 228 states that Table 3 presents land resources carriers, while the caption of Table 3 states these are loads. Table 2 caption is below the table.

References should be better formatted and line spacing reduced.

Author Response

Reviewer 1

(1) AHP should be better explained to those that are not familiar with this method.

Thanks for reviewer’s comments. The authors consider that AHP is just used to determine the weight of , , and  to respectively calculate ,, and . On the other hand, AHP is used to determine the weight of ,, and  to calculate . The weight is not the core of this study and the authors consider space limitation. Therefore, the authors introduce AHP method by reference Saaty (1977) in Line 462. Saaty (1977) first proposed the method of AHP and introduced it in detail.

(2) Values obtained in Table 5 and Table 10 and their interpretation i.e. the meaning should be better explained near the results.

Thanks for reviewer’s comments. In actual, the authors have interpreted the values in Table 5 and Table 10 in third paragraph of Discussion.

(3) There are some typos such as line 33: industrial activities; On the other hand.

Thanks for reviewer’s comments. The authors have revised this sentence now in line 44.

(4) Line 228 states that Table 3 presents land resources carriers, while the caption of Table 3 states these are loads. Table 2 caption is below the table.

Thanks for reviewer’s comments. The authors have revised line 245 “land resources carriers” as “land resources loads”. On the other hand, the authors have put Table 2 caption above the table.

(5) References should be better formatted and line spacing reduced.

Thanks for the re viewer’s comments. The authors have formatted the references and reduced the line spacing.

Reviewer 2 Report

Presented manuscript concentrated on the development of a new method for assessing land resources carrying capacity (LRCC) by examining the relationship between pressures caused by human activities and the supply capacity of land resources.

The Authors reviewed the literature published to date on this issue and concluded that there is a need for a universal model which describe LRCC with the application of the “load-carrier” perspective method.

The theme and main ideas of the article are therefore important for the International Journal of Environmental Research and Public Health.

I appreciate the article, the issues, and the question posed and analyzed by the authors, the topic is very interesting and actual . However, I have the following comments: 

(1) The title of the article should be slightly revised to take into account that this is a case study, the authors themselves write in line 13-16.... so here we have an example of a case study from The Chongqing (the municipality of  China)... I think this should be specified in the title..or Abstract.

(2) The literature review section (Introduction section, from line 156-163) should be strengthened to clearly summarise what has been found and what has not, in order to demonstrate the contribution of this study to the development of this topic.

(3) Line 216 - the title of the table shall be moved above the table.

(4) Lines 252-254 - what prompted the Authors to choose this particular indicator from among all those mentioned, whether arbitrarily or on the basis of expert knowledge…

(5) Lines 355-256 - The Authors refer to equations but it is not clear from the paper whether they are their own or taken from other studies, there is no indication of sources.

(6) I think that for chapters 3.1 Indicators for measuring land resources carriers and 3.2 Indicators for measuring land resources loads it would also be good to prepare a table containing the final indicators chosen to be implemented in the model... this is well described in chapter 3.1 and 3.2 but a tabular form, as a summary, would be clearer than a rather extensive text….

(7) For equations 8-10, no explanation in the text for all the markings used in the formulae.

(8) Lines 458-472 -  very brief description of the AHP method used for weighting which is very important here, it would be useful to have a graphical diagram for this part of the methodology…

(9) Figure 2 - in the legend there is information about the altitude range above sea level, this is missing in the legend that this is a numerical terrain model

(10) Lines 496-497 - What did the authors mean when they wrote about these supplements and the need to check…,  I do not see Supplementary data, no link..

(11) Table 4 - some descriptive data are aggregated to the county or municipality or city level, the Authors don’t write in the methodology how they aggregated all data to the county level…

(12) Table 4- I have a doubt about some of the data type in Table 4, for which the data is aggregated in a grid (x,y), the Authors state that it is at the resolution of county level..., shouldn't it be the grid and its dimensions, have the authors moved from the grid to the county level? Was a GIS type program used for this purpose?

(13) Line 500- where can I find Supplementary  data for these analyses

(14) Chapter 5. Discussion - discussion is not scientific, no reference of the results obtained to the results of other researchers.

(15) Lines 603-607 - for other regions of China and the world, would the proposed indicators and loads need to be modified due to the specificities of the region, the country...?

(16) lines 603-607 - there is no indication for which specific administrations, this methodology is recommended, this would increase the value of the manuscript…

(17) Line 612-613 - no link to the files.

(18) References - be organised arrange in accordance with the journal's guidelines.

Author Response

Reviewer 2

(1) The title of the article should be slightly revised to take into account that this is a case study, the authors themselves write in lines 13-16.... so here we have an example of a case study from The Chongqing (the municipality of China)... I think this should be specified in the title..or Abstract.

Thanks for the reviewer’s comments. In actual, the authors consider that this study is not a case study, and this study is a method innovation. Therefore, the selection of a case or cases is random. If data is available, the authors maybe select more cases not just Chongqing.

(2) The literature review section (Introduction section, from lines 156-163) should be strengthened to clearly summarise what has been found and what has not, in order to demonstrate the contribution of this study to the development of this topic.

Thanks for the reviewer’s comments. The authors consider that the literature review section conclude the third-sixth paragraph in Introduction not just from lines 156-163. In actual, the authors have summarized what has been found in the third-fifth paragraph and what has not in the sixth paragraph.

(3) Line 216 - the title of the table shall be moved above the table.

Thanks for the reviewer’s comments. The title of the table has been moved above the table.

(4) Lines 252-254 - what prompted the Authors to choose this particular indicator from among all those mentioned, whether arbitrarily or on the basis of expert knowledge…

Thanks for the reviewer’s question. According to the report by Chinese Nutrition Society (2016), food supply was divided into four types including supplies of grains, animal foods, processed foods, as well as vegetables and fruits (now in lines 261-263). The authors want to choose all indicators mentioned above to assess food supply, but data is not available. Therefore, the authors refer to published articles (now in lines 263-269), and find that total grain yield can be selected to assess food supply.

(5) Lines 355-356 - The Authors refer to equations but it is not clear from the paper whether they are their own or taken from other studies, there is no indication of sources.

Thanks for the reviewer’s question. The equations (5)-(6) now in lines 364-365 are from our own by Addition and Subtraction Method. The authors have explained the indicator  is from other study by Natural Capital Project now in lines 374-375.

On the other hand, the authors have added the explanations now in lines 362-363.

(6) I think that for chapters 3.1 Indicators for measuring land resources carriers and 3.2 Indicators for measuring land resources loads it would also be good to prepare a table containing the final indicators chosen to be implemented in the model... this is well described in chapter 3.1 and 3.2 but a tabular form, as a summary, would be clearer than a rather extensive text….

Thanks for the reviewer’s comments. The authors have added Tables 4 and 5 respectively in chapters 3.1 and 3.2 to summarize the final indicators chosen to be implemented in the model. The detailed revision can be seen in the manuscript.

(7) For equations 8-10, no explanation in the text for all the markings used in the formulae.

Thanks for the reviewer’s question. The authors have explained , , , , , and  of equations 8-10 in chapters 3.1 and 3.2 now in Tables 4 and 5.

(8) Lines 458-472 - very brief description of the AHP method used for weighting which is very important here, it would be useful to have a graphical diagram for this part of the methodology…

Thanks for reviewer’s comments. The authors consider that AHP is just used to determine the weight of , , and  to respectively calculate ,, and . On the other hand, AHP is used to determine the weight of ,, and  to calculate . The weight is not the core of this study and the authors consider space limitation. Therefore, the authors introduce AHP method by reference Saaty (1977) in line 477. Saaty (1977) first proposed the method of AHP and introduced it in detail.

(9) Figure 2 - in the legend there is information about the altitude range above sea level, this is missing in the legend that this is a numerical terrain model

Thanks for reviewer’s comments. The authors have revised Figure 2 with adding the legend about the altitude range above sea level. The detailed revision can be seen in Figure 2.

(10) Lines 496-497 - What did the authors mean when they wrote about these supplements and the need to check…, I do not see Supplementary data, no link.

Thanks for reviewer’s question. We are sorry for this. Supplementary data maybe not uploaded by the authors. The authors have uploaded Supplementary data in Round 1.

(11) Table 4 - some descriptive data are aggregated to the county or municipality or city level, the Authors don’t write in the methodology how they aggregated all data to the county level…

Thanks for reviewer’s question.

In actual, how all data are aggregated to the county level is written in Supplementary data. The authors hope that the resolution of all data is county level, but data is not available. Therefore, the authors replace municipality or city level as county level (see Supplementary data such as the calculation of ).

(12) Table 4- I have a doubt about some of the data type in Table 4, for which the data is aggregated in a grid (x,y), the Authors state that it is at the resolution of county level..., shouldn't it be the grid and its dimensions, have the authors moved from the grid to the county level? Was a GIS type program used for this purpose?

Thanks for reviewer’s question. The authors have moved from the grid to the county level with a GIS type program (see the calculation of  in Supplementary data).

(13) Line 500- where can I find Supplementary data for these analyses

Thanks for reviewer’s question. We are sorry for this. Supplementary data maybe not uploaded by the authors. The authors have uploaded Supplementary data in Round 1.

(14) Chapter 5. Discussion - discussion is not scientific, no reference of the results obtained to the results of other researchers.

Thanks for reviewer’s question. The first paragraph is the innovation of a load-carrier perspective method for evaluating land resources carrying capacity. The second paragraph is the application of the introduced method. Therefore, these two paragraphs should not need the results of other researchers.

The third-fourth paragraphs are discussions about the results of Chongqing case through a load-carrier perspective method. The authors discuss the results by referring to Chongqing Statistical Bureau (2020) (in line 585) and Liu et al. (2017) (in line 614). But the authors do not know why reviewer say no reference.

(15) Lines 603-607 - for other regions of China and the world, would the proposed indicators and loads need to be modified due to the specificities of the region, the country...?

Thanks for reviewer’s question. For other regions of China, the proposed indicators don’t need to be modified. But for other regions of the world, the proposed indicators maybe need to be modified due to the specificities of the country.

(16) lines 603-607 - there is no indication for which specific administrations, this methodology is recommended, this would increase the value of the manuscript…

Thanks for reviewer’s comments. The authors have added an indication for which specific administrations in lines 603-607 (now in line 623).

(17) Line 612-613 - no link to the files.

Thanks for reviewer’s question. We are sorry for this. Supplementary data maybe not uploaded by the authors. The authors have uploaded Supplementary data in Round 1.

(18) References - be organised arrange in accordance with the journal's guidelines.

Thanks for the re viewer’s comments. The authors have organised the references in accordance with the journal’s guidelines. The detailed revision can be seen in the manuscript.